# Expression Pattern of the SARS-CoV-2 Entry Genes *ACE2* and *TMPRSS2* in the Respiratory Tract

**DOI:** 10.3390/v12101174

**Published:** 2020-10-16

**Authors:** Yichuan Liu, Hui-Qi Qu, Jingchun Qu, Lifeng Tian, Hakon Hakonarson

**Affiliations:** 1Center for Applied Genomics, The Children’s Hospital of Philadelphia, Philadelphia, PA 19104, USA; liuy5@email.chop.edu (Y.L.); quh@email.chop.edu (H.-Q.Q.); jingchun.qu789@gmail.com (J.Q.); TianL@email.chop.edu (L.T.); 2Department of Pediatrics, The Perelman School of Medicine, University of Pennsylvania, Philadelphia, PA 19104, USA; 3Division of Human Genetics, Children’s Hospital of Philadelphia, Philadelphia, PA 19104, USA; 4Division of Pulmonary Medicine Children’s Hospital of Philadelphia, Philadelphia, PA 19104, USA

**Keywords:** ACE2, COVID-19, SARS-CoV-2, TMPRSS2

## Abstract

To address the expression pattern of the SARS-CoV-2 receptor ACE2 and the viral priming protease TMPRSS2 in the respiratory tract, this study investigated RNA sequencing transcriptome profiling of samples of airway and oral mucosa. As shown, ACE2 has medium levels of expression in both small airway epithelium and masticatory mucosa, and high levels of expression in nasal epithelium. The expression of ACE2 is low in mucosal-associated invariant T (MAIT) cells and cannot be detected in alveolar macrophages. TMPRSS2 is highly expressed in small airway epithelium and nasal epithelium and has lower expression in masticatory mucosa. Our results provide the molecular basis that the nasal mucosa is the most susceptible locus in the respiratory tract for SARS-CoV-2 infection and consequently for subsequent droplet transmission and should be the focus for protection against SARS-CoV-2 infection.

## 1. Introduction

Concerning the pandemic of coronavirus disease 2019 (COVID-19), on 26 September 2020, it had been diagnosed in 32.6 million people globally, causing 990,000 deaths. COVID-19 is caused by infection with the severe acute respiratory syndrome coronavirus 2 (SARS-CoV-2). For this highly infectious and deadly disease, there is no effective antiviral treatment [1]. The human angiotensin-I-converting enzyme 2 (ACE2) has been suggested to serve as the receptor for the cell entry of SARS-CoV-2 to cause infection [2]. ACE2 is a member of the renin–angiotensin system (RAS), with the function of converting angiotensin II to angiotensin-(1-7) (with seven amino acids), and converting angiotensin I to angiotensin-(1-9) [3], thereby negatively regulating the effects of angiotensin-I-converting enzyme (ACE) and the RAS system. In addition to its critical roles in RAS, ACE2 binds the S1 domain of the SARS-CoV Spike (S) protein as the viral receptor, and accounts for the infection of SARS-CoV and syncytia formation [4]. The genome sequence of SARS-CoV-2 shows significant similarity (79%) to that of SARS-CoV, while its receptor-binding domain shows even higher similarity to that of SARS-CoV [2], further supporting ACE2 as the receptor of SARS-CoV-2. COVID-19 is a highly infectious respiratory disease with a basic reproduction number, R0 (95% CI), of 3.28 (1.4, 6.49) [5]. After binding with ACE2, SARS-CoV-2 priming by the serine protease encoded by the transmembrane serine protease 2 gene (*TMPRSS2*) is also required for the viral entry into host cells [6,7]. In addition, three other genes that may be involved in the SARS-CoV-2 infection were highlighted by a recent study, i.e., furin, the paired basic amino acid cleaving enzyme gene (*FURIN*); the gene encoding the proprotein convertase subtilisin/kexin 3 (PCSK3); the dipeptidyl peptidase 4 gene (*DPP4*); and the basigin (Ok blood group) gene (*BSG*) [8]. PCSK3 may activate the SARS-CoV-2 S protein when *TMPRSS2* has low expression [9]. DPP4 and BSG have the possibility of serving as the alternative receptor of SARS-CoV-2 invasion [8].

Knowledge about the expression of *ACE2* and *TMPRSS2*, as well as other potential entry genes of SARS-CoV-2, namely *FURIN*, *DPP4*, and *BSG*, is extremely important to understand the infection of SARS-CoV-2 and to find ways to prevent the infection. For this purpose, we investigated RNA sequencing transcriptome profiling of samples of airway and oral mucosa, including small airway epithelium, alveolar macrophages, nasal epithelium, and masticatory mucosa. In addition, considering the critical role of mucosal-associated invariant T (MAIT) cells in mucosal immune defense against viral infection [10], transcriptome profiling of MAIT was also examined in this study.

## 2. Materials and Methods

Five datasets of transcriptome profiling by RNA sequencing (RNAseq) were acquired from the NCBI Gene Expression Omnibus (GEO) database (Table 1). We mapped and quantified the trimmed RNAseq reads using HISAT2 (https://ccb.jhu.edu/software/hisat2/index.shtml) to hg19 refSeq for each sample at default thresholds. The expression matrix was generated based on Cuffnorm functions in Cufflink package Version 2.2.1 [11]. Library sizes (i.e., sequencing depths) were normalized by the classic-fpkm method. Comparisons of the levels of *ACE*, *ACE2*, and *TMPRSS2* across different samples were based on the control or pre-exposure samples in each dataset, i.e., small airway epithelium of 10 healthy never-smokers before smoking E-cigarettes; alveolar macrophages of 10 healthy never-smokers before smoking E-cigarettes; nasal epithelium of 4 nonsmoker females before exposure to third-hand smoke; masticatory mucosa of 21 never-smokers; and MAIT of 5 healthy-bodyweight donors. The relative levels of the target genes were presented as the fragments per kilobase of transcript per million mapped reads (FPKM). Different types of human tissues were compared in this study. The expression of a single housekeeping gene (HKG) may not be constant in different cell types or different stages of the cell cycle. Instead, we chose 6 most stable HKGs with different essential functions for cell survival, to correct the expression levels of target genes. All values of the target genes were corrected by the average of relative levels of 6 HKGs, i.e., *ACTB*, *GAPDH*, *HMBS*, *HPRT1*, *RPL13A*, and *TBP*.

## 3. Results and Discussion

Gene expression patterns in five types of normal tissues are shown in Figure 1. As the major counterpart of *ACE2* expression, the expression of *ACE* is also investigated (Figure 1b). *ACE* and *ACE2* have medium and comparable levels of expression in both small airway epithelium and masticatory mucosa. These findings suggest that SARS-CoV-2 can infect both small airway epithelium and oral mucosa. The *ACE2* expression with the highest level of *TMPRSS2* expression in small airway epithelium provides explanation for the vulnerability infected individuals have for the characteristic pneumonia of COVID-19. The *ACE2* expression in masticatory mucosa helps explain the high level of infectivity via droplet transmission from SARS-CoV-2 infection residing in the oral mucous membrane. In addition to this study, the analysis on single-cell RNAseq (scRNAseq) data of human tissues showed high expression of *ACE2* in pulmonary type II alveolar cells (AT2) and respiratory epithelial cells [14]. Liao et al. further demonstrated expression of *ACE2* and its binding with the SARS-CoV-2 S protein in human bronchial epithelial cells [15]. In addition, a number of studies demonstrated the expression of *ACE2* in airway epithelial cells and its correlation with risk factors of severe COVID-19 [16,17].

Interestingly, the expression level of both *ACE2* and *TMPRSS2* in nasal epithelium is much higher than the levels of *ACE* expression, which is consistent with the studies by Sungnak et al. [18] and Bunyavanich et al. [19]. These results provide mechanistic evidence that the SARS-CoV-2 virus resides in both the oral and nasal mucosa of the upper respiratory tract, where it is able to bind to the *ACE2* receptor, serving as the principal locus of infections. These results also provide explanation for the high level of viral load in the oral and nasal mucosa and resulting high level of droplet transmission, with the lower airways being responsible for the severe form of pneumonia as well as the aerosol transmission of the virus. With *ACE2* and *TMPRSS2* as the major entry genes, however, other potential entry genes of SARS-CoV-2, namely *FURIN*, *DPP4*, and *BSG*, in the respiratory tract should not be neglected for the possibility of serving as the alternative pathway of SARS-CoV-2 infection.

The expressions of *ACE*, *ACE2*, and *TMPRSS2* are low in MAIT cells. The expression of *ACE* is high in alveolar macrophages, but the expression of *ACE2* cannot be detected with a low level of *TMPRSS2* in alveolar macrophages. These patterns of gene expression in the two types of innate immune cells suggest that SARS-CoV-2 has no or little direct impact on these two components of the innate immune system. In addition, these findings suggest that the infection of SARS-CoV-2 can be limited to the respiratory tract, which explains the absent of viremia in many COVID-19 patients [20,21].

In parallel, we examined whether the exposure factor in each RNAseq dataset affected the expression levels of *ACE*, *ACE2*, and *TMPRSS2*, to assess whether those common factors (i.e., smoking E-cigarettes, third-hand smoke, smoking, and obesity) are associated with the susceptibility of SARS-CoV-2 infection. FPKM values of *ACE*, *ACE2*, and *TMPRSS2* within each dataset were compared by paired T test (for small airway epithelium and nasal epithelium) or independent T test (for masticatory mucosa and MAIT). However, there was no difference observed (*p* > 0.05) between the dataset for *ACE*, *ACE2*, or *TMPRSS2*. Accordingly, while the SARS-CoV-2 virus is highly infectious, our results do not suggest a significant change in susceptibility to SARS-CoV-2 infection due to these factors. However, because of the modest sample size for each dataset, we must acknowledge that we are underpowered to identify minor effects.

In summary, this study highlights that the nasal mucosa is the most susceptible locus in the respiratory tract for SARS-CoV-2 infection and replication, is responsible for the subsequent high level of droplet transmission, and should be the focus for protection against SARS-CoV-2 infection, in line with a recent virological analysis [22]. Accordingly, local interventions with *ACE2* inhibitors [23] or *TMPRSS2* inhibitors (e.g., camostat mesylate) [7] may represent novel interventions to block SARS-CoV-2 cell entry and treat COVID-19.

## Figures and Tables

**Figure 1 viruses-12-01174-f001:**
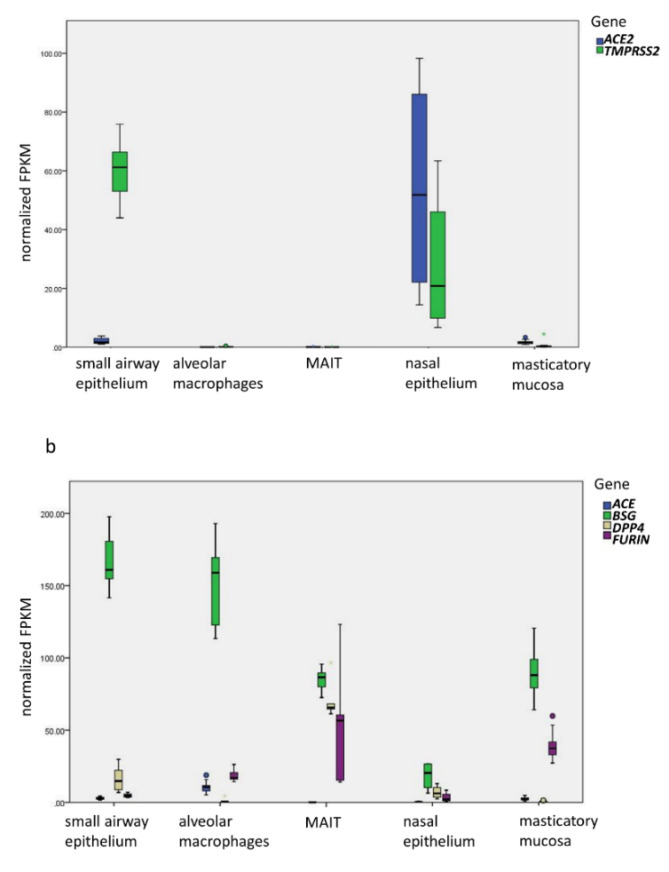
The expression of SARS-CoV-2 entry genes. (**a**) Established SARS-CoV-2 entry genes, *ACE2* and *TMPRSS2*, in five different types of samples. The difference between *ACE2* and *TMPRSS2* in the small airway epithelium is highly significant (*p* = 1.00 × 10^-12^). In the nasal epithelium, the expression of *ACE2* is significantly higher than that in small airway epithelium (*p* = 6.92 × 10^−4^), alveolar macrophages (*p* = 5.02 × 10^−4^), MAIT (*p* = 0.016), and masticatory mucosa (*p* = 5.48 × 10^−7^); the expression of *TMPRSS2* is lower than that in small airway epithelium (*p* = 5.20 × 10^−3^), but higher than in alveolar macrophages (*p* = 2.98 × 10^−3^), MAIT (*p* = 0.040), and masticatory mucosa (*p* = 1.40 × 10^−5^). (**b**) The expression of *ACE* and possible SARS-CoV-2 alternative entry genes, *BSG*, *DPP4*, and *FURIN*. Y-axis represents normalized FPKM values by the average of relative levels of 6 HKGs. The boxplot produced by the IBM SPSS Statistics Version 23 shows the mean, the first quartile and the third quartile, and the 95% confidence interval (CI). FPKM: fragments per kilobase of transcript per million mapped reads.

**Table 1 viruses-12-01174-t001:** Five datasets of RNAseq profiling analyzed in this study.

GEO Accession	Sample *	Data Description	Library Prep	Sequencing	Spots (M)	Bases	Size	GC Content	Reference
GSE85121	small airway epithelium [12]	10 healthy never-smokers before and after smoking E-cigarettes	TruSeq v2	Illumina HiSeq2500	39.1	9.8 Gbp	3.7 G	44.27%	hg19
GSE85121	alveolar macrophages [12]	10 healthy never-smokers before and after smoking E-cigarettes	TruSeq v2	Illumina HiSeq2500	37.5	9.4 Gbp	3.6 G	45.62%	hg19
GSE129959	nasal epithelium	4 nonsmoker females before and after exposure to third-hand smoke	Nextera XT	Illumina NextSeq500	51.1	4.2 Gbp	1.6 G	43.39%	hg19
GSE136262	masticatory mucosa [13]	21 never-smokers; 17 current smokers	TruSeq	Illumina HiSeq3000	16.5	840.4 Mbp	305.7 M	51.43%	hg19
GSE126169	MAIT [10]	5 healthy-bodyweight donors; 4 morbidly obese donors	SMART-Seq v4	Illumina NextSeq 500	25.2	2.1 Gbp	794.4 M	47.29%	hg19

* Includes the references showing the original source. MAIT: mucosal-associated invariant T (cells).

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
