# Peer review of "Expression Pattern of the SARS-CoV-2 Entry Genes ACE2 and TMPRSS2 in the Respiratory Tract"

_viruses, 2020, doi:10.3390/v12101174_

Round 1

Reviewer 1 Report

The manuscript has been improved and warrants publication.

Author Response

Thank you for your kind comments.

Reviewer 2 Report

It is glad that the authors improved the manuscript a lot with the first revision. I have additional comments for minor revision.

  1. For Table 1, what do the numbers in the parentheses mean in the "Sample" column? It appears to indicate the number of samples, but the numbers do not match the number in "Data description" column. Please describe the meanings of the numbers in footnote of the Table 1.
  2. The authors investigated the ACE expression levels as the counterpart of ACE2 gene. However, ACE is not the SARS-CoV-2 entry gene, unlike ACE2. Nonetheless, the authors describe the Figure 1a as "Established SARS-CoV-2 receptor genes" including ACE. This is incorrect. I suggest the authors to remove the data on ACE or to replace the ACE data to Figure 1b.
  3. For Figure 1, please indicate the Y-axis label within the figure itself with appropriate unit. It is recommended that the authors describe the values as "normalized FPKM" rather than naive "FPKM", since the correction process has been taken using HKGs as the authors denoted. Currently, no explanation within the Figure 1 or legend mention the correction process.
  4. In lines 121-122, the authors mentioned that "there was no difference observed (P>0.05) between the dataset for ACE, ACE2, or TMPRSS2." The authors mentioned GSE dataset numbers rather than anatomical sites. I recommend the authors to replace GSE85121, GSE129959, GSE136262, GSE126169 as alveolar macrophages, nasal epithelium, masticatory mucosa, MAIT, respectively. Additionally, p-value for difference between ACE2 and TMPRSS2 within the small airway epithelium (dataset GSE85121) appear to be significant based on Figure 1a. Please provide the p-value and indicate the statistical significance.
  5. The major novel result of this study is the highest expression levels of ACE2 and TMPRSS2 in nasal epithelium compared to other sites in the respiratory tract. Please indicate the p-values and statistical significance across the sample types of 6 genes investigated in this study.

Author Response

It is glad that the authors improved the manuscript a lot with the first revision. I have additional comments for minor revision.

Response: We thank you kindly for your instructive comments on our manuscript. We revised the paper carefully according to your comments. 

  1. For Table 1, what do the numbers in the parentheses mean in the "Sample" column? It appears to indicate the number of samples, but the numbers do not match the number in "Data description" column. Please describe the meanings of the numbers in footnote of the Table 1.

Response: We apologize for the confusion. The numbers in the parentheses mean the references/citations. The sample sizes are shown in the next column “data description”. We have added a footnote to explain this by your comment.

  1. The authors investigated the ACE expression levels as the counterpart of ACE2 gene. However, ACE is not the SARS-CoV-2 entry gene, unlike ACE2. Nonetheless, the authors describe the Figure 1a as "Established SARS-CoV-2 receptor genes" including ACE. This is incorrect. I suggest the authors to remove the data on ACE or to replace the ACE data to Figure 1b.

Response: We thank you for your instructive comments. We have carefully corrected the figure by your comment. In this revised version, the ACE2 gene is moved to Figure 1b.

  1. For Figure 1, please indicate the Y-axis label within the figure itself with appropriate unit. It is recommended that the authors describe the values as "normalized FPKM" rather than naive "FPKM", since the correction process has been taken using HKGs as the authors denoted. Currently, no explanation within the Figure 1 or legend mention the correction process.

Response: We thank you for your instructive comments. We have carefully added the description to the figure by your comment. The legend is amended for the correction process by your comment.

  1. In lines 121-122, the authors mentioned that "there was no difference observed (P>0.05) between the dataset for ACE, ACE2, or TMPRSS2." The authors mentioned GSE dataset numbers rather than anatomical sites. I recommend the authors to replace GSE85121, GSE129959, GSE136262, GSE126169 as alveolar macrophages, nasal epithelium, masticatory mucosa, MAIT, respectively. Additionally, p-value for difference between ACE2 and TMPRSS2 within the small airway epithelium (dataset GSE85121) appear to be significant based on Figure 1a. Please provide the p-value and indicate the statistical significance.

Response: We thank you for your instructive comments. We have carefully made the corrections by your comment. The significance P value is added to the figure legend.

  1. The major novel result of this study is the highest expression levels of ACE2 and TMPRSS2 in nasal epithelium compared to other sites in the respiratory tract. Please indicate the p-values and statistical significance across the sample types of 6 genes investigated in this study.

Response: We thank you for your instructive comments. We have carefully added the p-values to the figure legend by your comment.

Reviewer 3 Report

The authors modified the manuscript appropriately

Author Response

Thank you for your kind comments.